# Systemic Therapies for HER2-Positive Advanced Breast Cancer

**DOI:** 10.3390/cancers16010023

**Published:** 2023-12-20

**Authors:** Vasileios Angelis, Alicia F. C. Okines

**Affiliations:** Department of Medicine, Royal Marsden NHS Foundation Trust, Fulham Road, Chelsea, London SW3 6JJ, UK; v.angelis@nhs.net

**Keywords:** advanced breast cancer, antibody-drug conjugates, central nervous system, HER2-positive, resistance, tyrosine kinase inhibitors

## Abstract

**Simple Summary:**

HER2-positive breast cancer is an aggressive subtype of breast cancer which used to be associated with a poor prognosis, however this has been transformed by treatment advances over the past two decades. Patients with HER2-positive breast cancer are normally treated with a combination of two or more chemotherapy drugs and targeted agents which block the HER2 receptor, followed by surgery, radiotherapy and sometimes hormonal therapies. However, if the cancer is too extensive to be removed by surgery and/or has spread to other sites in the body, it is known as HER2-positive advanced breast cancer and regrettably is normally incurable. In this situation, sequential drug treatments, aiming to control the cancer for as long as possible and maintain quality of life are advised. The anti-HER2 drug, trastuzumab, was the first targeted agent developed for HER2-positive breast cancer and remains a key component of treatment for both early, curable and advanced incurable disease. Progress including the development of new targeted agents which synergise with trastuzumab and novel antibody-drug conjugates where chemotherapy is bound to the trastuzumab allowing selective delivery to cancer cells are discussed in this review.

**Abstract:**

Despite recent advances, HER2-positive advanced breast cancer (ABC) remains a largely incurable disease, with resistance to conventional anti-HER2 drugs ultimately unavoidable for all but a small minority of patients who achieve an enduring remission and possibly cure. Over the past two decades, significant advances in our understanding of the underlying molecular mechanisms of HER2-driven oncogenesis have translated into pharmaceutical advances, with the developing of increasingly sophisticated therapies directed against HER2. These include novel, more potent selective HER2 tyrosine kinase inhibitors (TKIs); new anti-HER2 antibody-drug conjugates; and dual epitope targeting antibodies, with more advanced pharmacological properties and higher affinity. With the introduction of adjuvant T-DM1 for incomplete responders to neoadjuvant therapy, fewer patients are relapsing, but for those who do relapse, disease that may be resistant to standard first- and second-line therapies requires new approaches. Furthermore, the risk of CNS relapse has not been abrogated by current (neo)adjuvant strategies; therefore, current research efforts are being directed towards this challenging site of metastatic disease. In this article, we review the currently available clinical data informing the effective management of HER2-positive breast cancer beyond standard first-line therapy with pertuzumab, trastuzumab, and taxanes, and the management of relapse in patients who have already been exposed to both these agents and T-DM1 for early breast cancer (EBC). We additionally discuss novel anti-HER2 targeted agents and combinations in clinical trials, which may be integrated into standard treatment paradigms in the future.

## 1. Introduction

HER2 protein overexpression and/or *HER2* gene amplification is present in approximately 15–20% of breast cancers and was associated with more aggressive disease and worse clinical outcomes [1,2]. The discovery of the HER2 receptor and advent of the HER2-directed monoclonal antibody (mAb) trastuzumab resulted in improved patient survival and has radically modified the natural history of HER2-positive breast cancer [3]. The treatment landscape of both early and advanced HER2-positive breast cancer has extensively changed over the past 20 years, with a rapidly expanding number of anti-HER2 therapeutics, including increasingly selective small molecule inhibitors, potent antibody-drug conjugates (ADCs), and bispecific antibodies, which have enriched our drug armamentarium and improved the survival for patients with HER2-positive advanced breast cancer (ABC). Despite these significant improvements in overall survival (OS), the majority of patients with HER2-positive ABC will ultimately die of this disease.

One of the greatest challenges in the treatment of HER2-positive metastatic breast cancer is that of primary and acquired resistance to anti-HER2 treatment. It is now recognised that a wide range of mechanisms of resistance may co-exist in the same patient [4]. This has become especially pertinent for patients relapsing after multi-agent (neo)adjuvant therapy, who may relapse with disease resistant to taxanes, trastuzumab, pertuzumab, and T-DM1.

The second major challenge is that of central nervous system (CNS) spread of HER2-positive ABC. Despite improvements in the management of early HER2-positive breast cancer, the rate of CNS relapses has not reduced. Furthermore, up to 50% of patients with HER2-positive ABC will develop brain metastases during the course of their disease, showing the need for better options for the prevention and treatment of both parenchymal metastases and the less common but devastating complication of leptomeningeal disease (LMD) [5,6].

Whilst not a systematic review, this article aims to provide an insight into the current anti-HER2 therapies and discuss the optimal sequencing in order to maximise the benefit of treatment and control of metastatic disease burden for patients with HER2-positive ABC.

## 2. First-Line Therapy: Trastuzumab, Pertuzumab, and Cytotoxic Chemotherapy

The combination of trastuzumab, pertuzumab, and a taxane is the standard first-line treatment for most patients with HER2-positive ABC. Trastuzumab is a humanised mAb (IgG1) that binds to the extracellular domain IV of HER2, whilst pertuzumab is a humanised mAb (IgG1) that binds to the extracellular domain II, preventing the formation of homo- and heterodimers and blocking one of the most powerful heterodimers, HER2/HER3, known to be the most potent activator of the PI3K/AKT signalling cascades (Figure 1) [7,8].

The approval of pertuzumab combined with trastuzumab and docetaxel in HER2-positive ABC is based on the results of the double-blinded, placebo-controlled phase III CLEOPATRA (Clinical Evaluation of Pertuzumab and Trastuzumab) trial [3] (Table 1). In this trial, 808 treatment-naïve patients with HER2-positive ABC were randomised to trastuzumab and docetaxel plus pertuzumab or placebo. Of note, less than half of the patients had received adjuvant or neoadjuvant chemotherapy, and only 11% had received (neo)adjuvant trastuzumab. The study met its primary endpoint, demonstrating a highly significant improvement in median progression-free survival (PFS) as well as overall response rate (ORR) and OS, as summarised in Table 1 [3,9]. A recent update of the study indicated that 37% of the patients were still alive after 8 years of follow-up [10], suggesting that a proportion of patients will experience long-term remission and potentially even cure with the triplet. The phase IIIb PERUSE trial usefully demonstrated that weekly paclitaxel substituted for docetaxel within this regimen has very similar outcomes, allowing use of the better-tolerated chemotherapy backbone [11]. Of interest, the benefit of adding pertuzumab to trastuzumab appears to be abrogated in the second line setting in combination with capecitabine, as the randomised phase II PHEREXA study failed to meet its primary endpoint of improved median PFS [12]. For patients relapsing within a year of neoadjuvant chemotherapy/dual anti-HER2 blockade, the chance of clinical benefit from retreatment with the same regimen is low, and it is not routinely recommended [13]. However, late relapses (beyond 12 months) may imply incomplete resistance, therefore rechallenge can be usefully attempted in this setting. A biopsy to confirm unchanged breast cancer phenotype is particularly important for such patients.

Patients with oestrogen receptor (ER)-positive, HER2-positive ABC can also be offered endocrine therapy combined with HER2-directed therapy as first-line treatment of metastatic disease in patients wishing to avoid or defer exposure to chemotherapy, although rapidly progressive disease or visceral crisis are contraindications to that approach. Two phase III studies demonstrated modest benefit from the addition of either trastuzumab [14] or lapatinib to endocrine therapy with an aromatase inhibitor (AI) [15], but the triplet combination of lapatinib, trastuzumab, and AI is more effective than either doublet in patients who have received prior chemotherapy and trastuzumab [16]. Furthermore, similar outcomes to those reported in the CLEOPATRA trial have been recently presented for the combination of pertuzumab, trastuzumab, and AI as first-line therapy for HER2-positive ABC, or, given as maintenance after prior chemotherapy, trastuzumab and pertuzumab in the phase II PERTAIN trial [17]. Most clinicians would favour the introduction of endocrine therapy in parallel with maintenance dual anti-HER2 antibodies following induction chemotherapy with a taxane, except for patients who are unsuitable to receive chemotherapy.

## 3. Second-Line Therapy: Trastuzumab Deruxtecan (T-DXd)

Trastuzumab-deruxtecan (T-DXd) is an ADC comprised of trastuzumab, a cleavable tetrapeptide drug linker, and a cytotoxic topoisomerase I payload, with a high drug-to-antibody ratio (ADR) of approximately 8:1 [18]. The payload of T-DXd easily crosses the cell membrane and therefore has a more potent cytotoxic effect on neighbouring cells, referred to as the bystander effect, whilst its short half-life ensures minimal systemic exposure [19,20].

In the initial phase I study of T-DXd, 118 patients with HER2-positive advanced breast cancer pre-treated with T-DM1 and a median of six prior lines of treatment were enrolled. An impressive ORR of 59.5% (95%CI, 49.7–68.7) was reported in the 111 evaluable patients, with a median response duration of 20.7 months [21]. The open-label, single-group, multi-centre, phase II DESTINY BREAST 01 study evaluated T-DXd in 184 patients with HER2-positive ABC with previous treatment with trastuzumab and T-DM1 and a median of six prior lines of therapy, and confirmed the remarkable activity observed in phase I [22]. T-DXd demonstrated durable activity with an ORR of 61.4% (95%CI 54–68.5), median PFS of 19.4 months (95%CI, 14.1–not reached), and median OS of 24.6 months (95%CI 23.1 months-not reached). In terms of tolerability and safety, common adverse events of grade ≥3 included neutropenia (20.7%), nausea (7.6%), and anaemia (8.7%). The most concerning toxicity observed was interstitial lung disease (ILD), which was grade 1–2 in the majority of cases (10.9%), but grade 3–4 in 0.5%, and fatal in 2.2%. T-DXd promptly received FDA breakthrough therapy and EMA approvals for patients with HER2-positive, locally advanced, or metastatic breast cancer who have been treated with trastuzumab and pertuzumab and progressed after T-DM1.

DESTINY-BREAST03 directly compared T-DXd to the previous standard second-line therapy, T-DM1, in a landmark randomised, open-label study for patients with HER2-positive, metastatic breast cancer previously treated with trastuzumab and taxane, bringing this agent into the second line setting (Table 1). Superiority of the newer ADC was clearly demonstrated in terms of median PFS (HR 0.33, 95%CI 0.26–0.43), ORR (79.7% vs. 34.2%), and overall survival (HR 0.64, 95%CI 0.47–0.87), establishing T-DXd as the standard of care [23,24]. Importantly, although ILD events were reported in 15% of participants, no cases of fatal ILD were reported in this study, demonstrating that careful patient selection, monitoring, and early introduction of steroids can prevent grade 5 ILD events from occurring.

The DESTINY-BREAST02 study compared T-DXd to capecitabine with the investigator’s choice of either trastuzumab or lapatinib, for patients with HER2-positive unresectable or metastatic breast cancer pre-treated with T-DM1 [25]. Again, the study demonstrated clear benefit from the ADC, with an ORR of 70% compared to 29% with the standard arm, median PFS 17.8 compared to 6.9 months, and median OS 39.2 compared to 26.5 months, confirming that this agent remains useful in later lines of therapy also.

The CNS penetration of T-DXd was initially uncertain due to the exclusion of patients with active brain metastases from the DESTINY breast studies. However, accumulating evidence supports this agent being highly effective for parenchymal brain metastases and potentially even leptomeningeal disease (LMD). The DESTINY-BREAST01 study included 24 patients with stable, treated brain metastases, in whom similar efficacy was seen to the overall study population (RR 58.3%, median PFS 18.1 months) [26]. Similarly, DESTNY-BREAST-03 included 114 patients with stable brain metastases in whom the benefit from T-DXd over TDM1 was maintained. The phase II TUXEDO-1 trial enrolled patients with untreated or progressive brain metastases and reported an impressive intracranial response rate of 73.3%, including two complete responses [27]. DESTINY-BREAST12 is a planned single-arm phase IV study that will evaluate T-DXd in patients with and without brain metastases [28]. Additionally, a currently recruiting phase II study (DEBBRAH) will investigate the CNS efficacy of T-DXd, including patients with leptomeningeal carcinomatosis as well as parenchymal disease. An interim analysis of 21 patients reported an encouraging intracranial response rate of 46.2% [29]. A case series of T-DXd in patients with LMD suggests promising activity in this poor-prognosis disease site [30].

## 4. Other Second-Line Therapy Options: Trastuzumab Emtansine (T-DM1)

Trastuzumab emtansine (T-DM1) is an antibody-drug conjugate (ADC) combining trastuzumab and the microtubule inhibitor DM1 via a thioether linker [21,31]. Preliminary phase I and II studies of T-DM1 in heavily pre-treated patients demonstrated meaningful activity with a favourable toxicity profile, which was later confirmed in the phase III TH3RESA trial [32,33,34,35,36,37]. Regulatory approval for T-DM1 was based on the pivotal EMILIA study, which randomised 991 patients with previously treated (taxane plus trastuzumab) locally advanced unresectable or metastatic HER2-positive breast cancer to T-DM1, or lapatinib plus capecitabine, the standard of care at that time. T-DM1 significantly improved both median PFS and OS [36,38] and was for many years the standard second-line therapy for HER2-positive ABC (Table 1). For patients relapsing soon after adjuvant TDM1, benefit from rechallenge is unlikely. However, for patients who have not received prior adjuvant TDM1 in whom T-DXD is contraindicated or unavailable, this remains a well-tolerated and effective option in the second line setting or beyond. Activity after prior T-DXd is unknown, and the combination of TDM1 with other agents has not proven successful thus far. The clinical activity of TDM-1 in combination with pertuzumab was compared to trastuzumab plus a taxane (docetaxel or paclitaxel) or TDM1 alone in the first-line phase III MARIANNE trial [39]. The median OS was similar in the three arms, demonstrating no significant benefit from combining T-DM1 with pertuzumab, a finding replicated in the neoadjuvant setting [40].

## 5. Third Line Options: Tyrosine Kinase Inhibitors

Tyrosine kinase inhibitors (TKIs) are orally bioavailable small molecules with a lower molecular weight than monoclonal antibodies, thus allowing effective penetration through the blood–brain barrier and useful efficacy in CNS metastatic disease (Table 2).

### 5.1. Lapatinib

Lapatinib is a dual small-molecule inhibitor of both HER2 and the epidermal growth factor receptor (EGFR) and the first TKI approved in HER2-positive ABC. Following pre-clinical studies confirming synergy between lapatinib and trastuzumab [41], this chemotherapy-free combination was evaluated in the EGF104900 study, which demonstrated an OS benefit in heavily pre-treated patients compared with lapatinib alone [42,43]. Lapatinib was approved by the FDA in 2007 based on the pivotal phase III study by Geyer et al., where its addition to capecitabine led to a significantly improved time to progression (TTP: 8.4 months vs. 4.4 months, HR = 0.49, 95%CI 0.34–0.79, *p* < 0.001) and a trend towards improved median OS in the combination arm (75 weeks vs. 64.7 weeks; HR = 0.87, 95%CI 0.71–1.08, *p* = 0.210) [44]. Importantly, lapatinib was the first HER2-targeted agent to demonstrate activity in the treatment of CNS metastases and, although single agent activity is minimal (6% response rate in patients with progressive CNS disease post radiotherapy) [45], combination with capecitabine increased this to 31.8% [46]. Furthermore, for patients with previously untreated CNS metastases, an ORR of 65.9% and median OS of 17 months were reported in a phase II study of the doublet [47]. Regrettably, the phase III CEREBEL (EGF111438) was unable to demonstrate a preventative role, with no reduction in the development of CNS metastases between patients treated with lapatinib/capecitabine and those randomised to lapatinib/trastuzumab [48]. Of note, the rates in both arms of the study were unexpectedly low, at least in part due to baseline MRI screening. Following the results of the EMILIA trial, the use of lapatinib/capecitabine shifted to the post-TDM1 setting.

### 5.2. Neratinib

Neratinib (HKI-272) is a potent irreversible inhibitor of HER1, 2, and 4, which is highly active against HER2-overexpressing human breast cancer cell lines in vitro, with the potential to overcome resistance to trastuzumab in *HER2*-amplified breast cancer cells [49].

Neratinib was assessed in the first line metastatic setting in the phase II NEfERT trial, where 479 patients with HER2-positive ABC were randomised to receive paclitaxel plus either neratinib or trastuzumab. No improvement in median PFS was demonstrated (12.9 months, HR 1.02; 95%CI 0.81–1.27, *p* = 0.89), and neratinib was associated with much higher toxicity, and with grade 3 diarrhoea in up to 30% of patients [50]. However, a subsequent analysis indicated that incidence of CNS recurrences was significantly lower with neratinib (relative risk 0.48, 95%CI 0.29–0.79, *p* = 0.002), and time to CNS metastases was also significantly delayed (HR 0.45; 94% CI 0.26–0.78, *p* = 0.004) [51]. Similarly, in the phase II TBCRC 022 trial, conducted in patients with HER2-positive ABC with brain metastases, the combination of neratinib with capecitabine demonstrated an impressive CNS response rate of 49% [52]. This regimen was subsequently evaluated in the phase III NALA trial, which directly compared neratinib plus capecitabine vs. lapatinib plus capecitabine in patients with HER2-positive ABC who had received at least two previous lines of anti-HER2 treatments [53]. This reported an improvement in 12-month PFS in patients receiving neratinib plus capecitabine (28.8% vs. 14.8%; *p* = 0.059); however, this was at the cost of more toxicity, with 25% of patients on the neratinib combination experiencing grade 3 diarrhoea vs. 15% of patients on the lapatinib arm, despite anti-diarrhoeal prophylaxis with loperamide in the investigational arm. Importantly, the time to intervention for symptomatic CNS disease was significantly delayed with neratinib compared to lapatinib, and the overall incidence of CNS interventions required was lower with neratinib/capecitabine (22.8%) compared to lapatinib/capecitabine (29.2%). Furthermore, a recent subgroup analysis of the 101 patients with known baseline CNS involvement enrolled in the NALA trial confirmed fewer CNS interventions and prolonged CNS PFS with neratinib/capecitabine (7.8 months vs. 5.5 months, HR 0.66, 95%CI 0.59–1.38) compared to lapatinib plus capecitabine [54]. As such, this is an important and effective combination, including for patients with brain metastases, but can be challenging to deliver due to the EGFR-blockade related toxicities, in particular the high rate of grade 3 diarrhoea.

Neratinib is currently being evaluated in combination with other agents including T-DM1 and endocrine therapies. For example, a phase I/II trial evaluating the safety, tolerability, and efficacy of T-DM1 with neratinib in patients progressing after dual HER2 blockade with trastuzumab and pertuzumab indicated an impressive ORR of 64%, and the phase II extension of this study is ongoing [55].

### 5.3. Tucatinib

Tucatinib (ONT-380, ARRY-380) is a tyrosine kinase inhibitor that is highly selective for the kinase domain of HER2 and is approximately 1000-fold more potent for HER2 than EGFR [56]. Pre-clinical activity in xenograft models of CNS metastases [57] led to early evaluation in patients with breast cancer brain metastases, but only modest CNS activity was reported in the phase I dose escalation study of tucatinib plus trastuzumab [58], precipitating studies of chemotherapy combinations. In one of two parallel phase Ib trials of tucatinib combinations, 23 heavily pre-treated patients with HER2-positive ABC, including patients with brain metastases, were treated with tucatinib in combination with capecitabine and/or trastuzumab, with an encouraging ORR of 61% in the triplet combination arm and a median duration of response of 10 months (95%CI: 2.8–19.3) [59].

HER2CLIMB was a pivotal randomised phase II study, in which 612 patients with HER2-positive ABC, with or without CNS metastases, were assigned 2:1 to capecitabine and trastuzumab with the addition of tucatinib or placebo [60]. All patients had previously been treated with a taxane, trastuzumab, pertuzumab, and T-DM1. Of particular interest was the inclusion of patients with active (untreated or progressive) CNS metastases, who did not require immediate locoregional treatment. The one-year PFS rate was 33.1% in the tucatinib arm and 12.3% in the placebo group (HR 0.54, 95%CI 0.42–0.71 *p* < 0.001), meeting the primary endpoint of the study. For the subgroup of 291 patients with CNS metastases, median PFS was 7.6 months vs. 5.4 (HR 0.48, 95%CI, 0.34–0.69; *p* < 0.001). Importantly, OS was also improved with tucatinib (2-year OS 44.9% compared to 26.6% with placebo: HR, 0.66; 95%CI, 0.50–0.88; *p* = 0.005). The triplet regimen was well tolerated, with higher rates of any grade nausea and vomiting, and diarrhoea, but similar rates of grade 3 adverse events in both arms, with only grade 3 ALT/AST rises being significantly more common with tucatinib than placebo after adjustments for the longer duration of treatment exposure with tucatinib [61]. Based on the HER2-CLIMB results, tucatinib obtained FDA approval and EMA CHMP recommendation for approval for patients progressing after one or more prior anti-HER2-directed treatments. The triplet regimen is also under evaluation in patients with difficult-to-treat and poor-prognosis leptomeningeal disease (NCT03501979); in this study, tucatinib was confirmed in the CSF at similar levels to plasma, and analysis of 17 patients enrolled in TBCRC049 reported a median time to CNS progression of 6.9 months and median OS of 11.9 months [62,63].

The efficacy of tucatinib after T-DXd has not been evaluated and may depend upon the mechanism of resistance to T-DXd; translational results from the DAISY trial, which have been presented but not yet published, reported a reduction in HER2 expression in 65% of patients at progression on T-DXd, suggesting that re-biopsy to confirm HER2 overexpression may be warranted for patients not responding to anti-HER2 therapies after T-DXd. However, as TKIs are usually delivered with capecitabine, this could be a useful combination therapy post T-DXd as the cytotoxic effect may be active in any areas of HER2-negative disease that may have developed during T-DXd (which would likely exhibit resistance to the TKI).

The combination of tucatinib with other agents is being actively pursued, with a number of studies now ongoing, including HER2CLIMB-02 (NCT03975647), comparing tucatinib to placebo in combination with T-DM1, which recently reported a moderate improvement in median PFS with the addition of tucatinib (median 9.5 vs. 7.4 months, HR 0.76, 95%CI 0.61–0.95, *p* = 0.0163), but unfortunately coupled with a significant rate of grade ≥ 3 hepatotoxicity in 28.6% [64]. The phase 3 study was based on a previously reported phase Ib trial of tucatinib with T-DM1 in pre-treated patients with HER2-positive ABC, which demonstrated an acceptable safety profile and encouraging response rate (48%) and median PFS (8.2 months) in patients previously treated with trastuzumab plus a taxane [65]. The TOPAZ trial (NCT04512261) will evaluate tucatinib in combination with trastuzumab and the anti-PD-1 antibody pembrolizumab in patients with brain metastases. Perhaps most exciting, however, is the evaluation of tucatinib in combination with the antibody-drug conjugate trastuzumab deruxtecan in the HER2CLIMB-04 study (NCT04539938). Studies focusing on management of patients with CNS metastases are summarised in Table 3.

### 5.4. Novel Tyrosine Kinase Inhibitors

#### 5.4.1. Pyrotinib

This irreversible pan-HER TKI was evaluated in a phase I clinical trial with a phase II expansion arm, where Chinese patients with HER2-positive advanced breast cancer pre-treated with at least two lines of therapy were randomised to receive pyrotinib/capecitabine or lapatinib/capecitabine [66,67]. The pyrotinib combination arm demonstrated promising activity, with an ORR of 78.5% vs. 57.1% in the capecitabine/lapatinib arm, and a significant improvement in PFS (median 18 vs. 7 months, HR 0.36; *p* < 0.001). However, pyrotinib was associated with higher rates of grade 3 adverse events, which included palmo-plantar erythema (25 vs. 21%), diarrhoea (15 vs. 5%), and neutropenia (9 vs. 3%). An intriguing finding in the parallel biomarker analysis indicated that *PIK3CA* and *TP53* mutations in circulating tumour DNA may predict response to pyrotinib. The role of pyrotinib was also investigated in the phase III PHENIX trial, where 279 patients previously treated with trastuzumab and a taxane were randomised to receive capecitabine plus pyrotinib or placebo [68]. Median PFS was 11.1 with pyrotinib vs. 4.1 months with placebo (HR 0.18; 95%CI 0.12–0.26; *p* < 0.001), with an impressive ORR of 68.6% reported with pyrotinib plus capecitabine, vs. 16.0% with placebo/capecitabine. The lack of trastuzumab in the control arm as well as the inclusion of patients who had not received prior pertuzumab or T-DM1 prevents cross-trial comparison with other HER2 TKIs and inevitably limits the application of these results to clinical practice. In the recently published phase III PHOEBE trial, the combination of pyrotinib/capecitabine was compared to lapatinib/capecitabine in patients with HER2-positive ABC who had previously received trastuzumab and taxane chemotherapy and reported improved median PFS with pyrotinib (12.5 vs. 6.8 months; HR 0.39 (95%CI 0.27–0.56) *p* < 0.0001), but a much higher rate of grade 3–4 toxicities, with up to 30% of the ITT population experiencing grade 3–4 diarrhoea, similar to that expected with capecitabine/neratinib [69]. The CNS activity of pyrotinib was also investigated in combination with capecitabine in a recent phase II trial of 61 HER2-positive ABC patients with untreated brain metastases (PERMEATE); the objective CNS response rate (defined by RECIST 1.1 criteria) was 74.6% (95%CI 61.1–85.0) [70]. Overall, this can be concluded to be an active drug, including in brain metastases, but its clinical utility is currently limited by its significant toxicities and the uncertainty over its activity in ADC-negative pre-treated patients.

#### 5.4.2. Poziotinib

This novel potent irreversible pan-HER TKI showed potent activity when evaluated as a single agent in a phase II trial of 102 heavily pre-treated patients with HER2-positive ABC [71]. Despite a promising disease control rate of 75.5%, the median PFS was only 4 months (95%CI, 2.9–4.4 months). Furthermore, there were concerns regarding tolerability of this drug due to the high rate of grade 3 diarrhoea (38%) reported, amongst other toxicities. A further phase II study demonstrated similar efficacy and toxicity in a predominantly Caucasian population, with median PFS 4.1 and 5.8 months in intermittent and continuous dosing cohorts, respectively, but grade 3–4 diarrhoea in 30% of patients, suggesting that combination with a cytotoxic such as capecitabine could be extremely challenging [72]. A pre-clinical study suggested that poziotinib could be explored in the TKI-resistant setting, with partial regression seen in a cell line model of acquired *HER2 L755S* mutations that were resistant to lapatinib, neratinib, and tucatinib [73]. Clinical confirmation of this finding could usefully place this agent in the post-T-DXd/T-DM1/tucatinib setting. Novel agents in clinical trial are summarised in Table 4.

### 5.5. Trastuzumab Duocarmazine

Trastuzumab duocarmazine (SYD985) is an ADC comprised of trastuzumab conjugated via a cleavable linker with an alkylating-based pro-drug (secoduocarrmycin hydroxybenzamideazaindole, seco-DUBA) with a drug to antibody ratio of 2.7:1. Based on encouraging findings in early phase studies [74,75,76], the safety and activity of SYD985 was assessed in a phase I trial enriched for patients with HER2-positive advanced breast cancer. The most common reported grade ≥ 3 adverse events included neutropenia (6%) and conjunctivitis (4%). SDY985 was subsequently evaluated in the post-T-DM1 setting in the phase III trial TULIP study, in which patients were randomised between SYD985 vs. therapy of the physician’s choice (trastuzumab or lapatinib plus cytotoxic). The study met the primary endpoint of improved PFS (median 7 vs. 4.9 months, HR 0.64, 95%CI 0.49–0.84, *p* = 0.002), but the results were overshadowed by the greater benefit reported with T-DXd in the DESTINY-BREAST-03 trial, which was presented at the same conference. Furthermore, application of the results into clinical practice are limited by the frequent toxicity of conjunctivitis and keratitis in almost 40% of patients, as well as a low rate of ILD (7.6%), which included two grade 5 events [77].

### 5.6. ARX788

ARX788 is an anti-HER2 ADC covalently bound to a novel tubulin inhibitor payload, amberstatin (AS269), with a high ADR of 1:9. The phase I ACE-BREAST-01 trial evaluated this agent in patients with heavily pre-treated HER2-positive ABC. A median PFS of 17 months was reported in the 69 patients who had received a median of six prior lines of therapy [78,79]. The phase III ACE-BREAST-02 trial randomised 441 patients pre-treated with a taxane and trastuzumab to ARX788 or capecitabine and lapatinib, and it was confirmed that the study met its primary endpoint of superior PFS. Hopefully, future studies will confirm activity in patients who have already received standard therapies for HER2-positive ABC [80].

### 5.7. Other Therapies

#### 5.7.1. Margetuximab (MGAH22)

Margetuximab is an Fc-optimized chimeric monoclonal antibody derived from 4D5, a precursor of trastuzumab. The mechanism of action of margetuximab relies on its high affinity of binding to effector cells, which increases antibody-dependent cell-mediated cytotoxicity [81]. The phase III SOPHIA study evaluated the efficacy of margetuximab in combination with chemotherapy in 536 heavily pre-treated patients compared to trastuzumab plus chemotherapy [82]. Results indicated a modest improvement in PFS (5.8 vs. 4.9 months, HR 0.76; 95%CI, 0.59–0.98; *p* = 0.033) and a numerical but not statistically significant OS benefit (21.9 vs. 19.8 months, HR, 0.89; 95%CI, 0.69–1.13; *p* = 0.326). In a planned exploratory analysis by Fc-gamma receptor genotype, margetuximab significantly improved median PFS in CD16A-158F carriers (6.9 vs. 5.1 months; HR 0.68; *p* = 0.005). The SOPHIA study is one the first studies to represent the impact of Fc-gamma receptor genotype on the efficacy of anti-HER2 therapy, but it raises the challenging issue of implementing patient selection according to genotyping into routine clinical practice, when there is currently no companion diagnostic test for this agent.

#### 5.7.2. Bispecific Antibodies

Bispecific antibodies are monoclonal antibodies that combine the functional elements of two monoclonal antibodies binding on two different targets or epitopes, either in the same or in different receptors. Their mode of action thus involves interference with two or more tyrosine kinase signalling pathways, by targeting either the relevant receptor or its ligand. The rationale of developing bispecific antibodies is to target two epitopes on the same or distinct molecules, instead of using two separate monoclonal antibodies (such as trastuzumab and pertuzumab).

Zenocutuzumab (MCLA-128) is an IgG1 targeting both HER2 and HER3 with enhanced antibody-dependent cytotoxicity (ADCC), designed to overcome resistance that can occur secondary to the heterodimerization of HER2 and HER3 [83]. Results from a phase I/II study in solid tumours, including 11 patients with heavily pre-treated HER2-positive ABC, reported an encouraging clinical benefit rate of 64% [84]. Common adverse events included asthenia and fatigue, but there was no significant cardiotoxicity. A phase II study evaluating MCLA-128 in HER2-positive patients in combination with trastuzumab, with or without chemotherapy, and *PIK3CA* mutation in ER-positive/HER2-low patients has completed accrual, but results are awaited (ClinicalTrials.gov identifier: NCT03321981).

Zanidatamab (ZW25) is a novel bispecific antibody that binds to two non-overlapping HER2 epitopes. It is characterised by a high potential for immunomodulating activities and a more potent cytotoxic effect than trastuzumab, via blockade of ligand-dependent and independent growth [85]. Clinical benefit was reported in 54% of solid tumour patients in a phase I study evaluating the safety and efficacy of single-agent ZW25 [86]. Ongoing studies are evaluating its activity alone and combined with capecitabine, vinorelbine, or paclitaxel in patients with HER2-expressing ABC (ClinicalTrials.gov identifier: NCT02892123), which has completed accrual. The results of zanidatamab combined with palbociclib plus fulvestrant in patients with ER-positive HER2-positive ABC (ClinicalTrials.gov identifier: NCT04224272) have been presented but not published, reporting a median PFS of 11.7 months but 53% grade ≥ 3 neutropenia and 14% grade ≥ 3 diarrhoea [87].

#### 5.7.3. Bispecific Fusion Proteins

PRS-343 is a bispecific fusion protein targeting HER2 and CD137, a key member of the tumour necrosis factor receptor superfamily. Dual targeting of CD137 and HER aims to bridge CD137-expressing T-cells in close proximity to HER2-positive cancer cells, generating a strong signal to tumour antigen-specific T-cells, eliciting anti-tumour activity and thereby providing a potent costimulatory signal to tumour antigen-specific T-cells [88]. A phase I trial evaluating PRS-343 in HER2-positive solid tumours demonstrated encouraging evidence of safety and clinical benefit with a correlative pharmacodynamic effect in a heavily pre-treated population, the latter confirmed by CD8+ T cell immunohistochemistry on tumour biopsies pre and post treatment [89]. The results of a trial evaluating its role in combination with the anti-PDL-1 antibody atezolizumab are awaited (ClinicalTrials.gov identifier: NCT03650348).

#### 5.7.4. Immune Checkpoint Inhibitors

Like triple-negative breast cancer, HER2-positive breast cancer frequently has a high tumour mutational burden and high rate of tumour-infiltrating lymphocytes (TILs), and given this, patients have the potential to benefit from immunotherapy [90]. PD-L1 is also frequently expressed in HER2-positive breast cancer cells and is associated with favourable survival [91]. Preclinical studies further support the combination of checkpoint inhibitors with anti-HER2 therapies through the synergistic activation of CD81 T-cells [92,93].

##### Anti-PDL1 Agents

Studies evaluating anti-PDL1 agents in HER2-positive breast cancer have thus far reported unremarkable results. The phase I JAVELIN study reported a poor response rate (5.4%) to avelumab monotherapy in patients with heavily pre-treated HER2-positive breast cancer [94]. The phase II KATE2 trial evaluated the efficacy and safety of another anti-PDL1 agent, atezolizumab, in combination with T-DM1 in pre-treated patients with advanced HER2-positive breast cancer [95] but failed to demonstrate a clinically meaningful improvement in PFS (8.2 vs. 6.8 months; HR = 0.82, 95%CI 0.55–1.23, *p* = 0.33) and showed an association with more adverse events. On post hoc analysis, a non-statistically significant benefit was seen in the sub-group of patients with PDL-1-positive cancers (RR 54% vs. 33% and median PFS 8.5 vs. 4.1 months, HR 0.60, 95%CI 0.32–1.11). Based on these data and the higher likelihood of PDL-1 expression in EBC, the ASTEFANIA trial will assess the benefit of adding atezolizumab to adjuvant TDM1 in patients with high-risk EBC not achieving pCR after neoadjuvant chemotherapy, trastuzumab, and pertuzumab (NCT04873362). PDL1/PD1 inhibitors may have a different effect in EBC, as they result in improved pCR in early triple-negative breast cancer regardless of PDL1 status, so the outcome of this study is eagerly awaited. Nonetheless, the introduction of Enhertu in this setting may subsequently supersede this approach.

A planned phase Ib study of T-DXd combinations will include cohorts combining T-DXd with the anti-PDL-1 antibody durvalumab +/− paclitaxel (DESTINY BREAST07, NCT04538742), and the ongoing phase III NRG-BR004 trial is evaluating the combination of paclitaxel, trastuzumab, and pertuzumab, with or without atezolizumab, in the first line setting (ClinicalTrials.gov identifier: NCT03199885).

##### Anti-PD1 Agents

The first study to evaluate the addition of the anti-PD1 agent pembrolizumab to trastuzumab was the single-arm phase Ib/II PANACEA trial. This study, which did not select patients by PDL-1 status, demonstrated an ORR of 15.2% and a median of PFS and OS of 2.7 months and 16 months, respectively, in the PDL-1-positive population (*n* = 40), with no benefit in the PDL-1-negative sub-group (*n* = 12) [96]. Although these outcomes are not particularly encouraging, all patients had previously progressed on trastuzumab, so the responses in the PDL-1-positive subgroup warrant further research. A phase I study evaluating the combination of T-DM1 with pembrolizumab (ClinicalTrials.gov identifier: NCT03032107) has completed recruitment. A phase Ib study will also establish the maximum tolerated dose for the combination of pembrolizumab with T-DXd, with expansion cohorts in HER2-positive and HER2-low ABC (ETUDE, NCT04042701).

The combination of nivolumab and T-DXd is undergoing evaluation in a phase Ib study including patients with ABC with both high and low HER2 expression (NCT03523572). Preliminary data presented recently were disappointing compared to results with T-DXd monotherapy; whilst the ORR of 59.4% in the 32 HER2-positive patients was comparable to the monotherapy data, the median PFS reported was just 8.6 months [97]. Table 5 illustrates current studies assessing the combination of checkpoint inhibition with anti-HER2 therapies. For a more comprehensive review of the future of checkpoint inhibitors in HER2-positive breast cancer, see the review by Agostinetto et al. [98].

#### 5.7.5. CDK4/6 Inhibitors

The emergence of cyclin-dependent kinase 4 and 6 (CDK4/6) inhibitors has radically changed the landscape of treatment for ER-positive/HER2-negative ABC. The CDK4 and 6 pathways can mediate resistance to HER2-targeted agents, and the activity of CDK4/6 inhibitors on luminal HER2-expressing cells is well established [99], providing a basis for clinical evaluation of combinations of CDK4/6 inhibitors with HER2-targeted agents in ER-positive HER2-positive breast cancer. Of note, studies evaluating CDK4/6 inhibitors as monotherapy were attempted, such as the JPBO trial of abemaciclib, but no objective responses were seen at the time of the interim analysis, so the ER-positive, HER2-positive ABC cohort was not expanded [100].

MonarchHER was a randomised phase II trial comparing an abemaciclib and trastuzumab doublet, with and without fulvestrant, to chemotherapy of the physician’s choice plus trastuzumab in patients with ER-positive HER2-positive ABC [101]. This study demonstrated an improved ORR (35.4% vs. 22.8% with chemotherapy plus trastuzumab) and median PFS with the endocrine/targeted therapy triplet (8.3 vs. 5.7 months; HR = 0.673, 95%CI, 0.451–1.003; *p* = 0.0253) but no significant difference between the doublet and chemotherapy/trastuzumab arms. Regrettably, the study did not include a fulvestrant plus trastuzumab arm, thus complicating interpretation of the data with regards to the contribution of abemaciclib. However, the triplet regimen is a potential option for patients with ER-positive/HER2-positive disease after standard therapies have been exhausted, or for those preferring to avoid chemotherapy.

Similarly, the phase II SOLTI-1303 PATRICIA study evaluated the use of palbociclib and trastuzumab, with and without letrozole, in heavily pre-treated patients with HER2-positive advanced breast cancer. Of interest, the triplet combination was particularly active in the luminal A and B subtypes defined by PAM50, with a median PFS of 12.4 vs. 4.1 months (HR 0.30, 95%CI;0.11–0.86, *p* = 0.025) compared to non-luminal subtypes [102].

Tucatinib has also been assessed in combination with palbociclib and letrozole in a phase Ib/II trial patients with ER-positive HER2-positive ABC, demonstrating an acceptable safety profile and encouraging clinical activity in the phase I part of the study, but the results of the phase II part of the study are still awaited [103].

Other ongoing studies include the PATINA study (ClinicalTrials.gov identifier: NCT02947685), which will evaluate the combination of palbociclib with trastuzumab, pertuzumab, and an aromatase inhibitor after an induction of standard first-line therapy; and phase Ib/II studies of palbociclib and T-DM1 (ClinicalTrials.gov identifier: NCT01976169); palbociclib, trastuzumab, pertuzumab, and anastrozole (ClinicalTrials.gov identifier: NCT03304080); and ribociclib with trastuzumab or T-DM1 (ClinicalTrials.gov identifier: NCT02657343).

#### 5.7.6. PI3K/Akt/mTOR/AKT Inhibitors

The PI3K/AKT/mTOR pathway has been implicated in the development of resistance to endocrine, cytotoxic, and HER2-directed therapy in breast cancer [104]. Trastuzumab blocks the signalling pathway upstream of PI3K; hence, downstream dysregulation, such as PTEN loss or *PI3KCA* mutations can override HER2 receptor blockade. Inhibition of PI3K leads to enhanced HER2 signalling in HER2-overexpressing breast cancer cells, suggesting that inhibitors targeting this pathway may act synergistically with trastuzumab in cells demonstrating resistance [105].

#### 5.7.7. mTOR Inhibitors

Although early phase studies suggested that mTOR inhibitors combined with cytotoxic chemotherapy and trastuzumab had clinical activity in HER2-positive ABC, subsequent phase III studies demonstrated no clinically significant benefit [106,107]. In the BOLERO-1 trial, everolimus was assessed in combination with trastuzumab and paclitaxel in the first line setting. No improvement in PFS was reported from the addition of everolimus, and although there was a numerical benefit of 7 months in the sub-group of ER-negative patients, it did not reach statistical significance [108]. In the BOLERO-3 trial, the combination of everolimus with trastuzumab and vinorelbine was evaluated in patients with trastuzumab-resistant ABC, and a small but statistically significant benefit in PFS was reported: median PFS 7.0 vs. 5.78 months (HR = 0.78; 95%CI, 0.65–0.95; *p* = 0.0067) [109]. A combined exploratory biomarker analysis from both studies indicated that dysregulation in the PI3K pathway in the form of *PIK3CA* mutation and/or *PTEN* loss and/or *AKT1* mutation were associated with improved PFS, suggesting a possible benefit for patients molecularly selected for PI3K pathway aberrations [110].

#### 5.7.8. PI3K Inhibitors

Targeting upstream molecules in the PI3K/AKT/mTOR pathway has been challenging, and many of the PI3K and Akt inhibitors studied have not progressed beyond early phase trials. The alpha-specific PI3K inhibitor alpelisib demonstrated clinical activity in combination with T-DM1; in a phase I study that included heavily pre-treated HER2-positive patients, an ORR of 43% and a median PFS of 8.1 months (95%CI 3.9–10.8) were reported [111], although these are similar to the outcomes reported with T-DM1 alone in the EMILIA trial, both in the overall population and the sub-group of patients with *PIK3CA* mutations [112]. The most commonly reported adverse events included rash, gastrointestinal side effects, thrombocytopenia, anaemia, transaminitis, and hyperglycaemia. Notably, alpelisib demonstrated significant clinical activity in patients who had previously progressed on T-DM1, with an encouraging ORR of 30% and median PFS of 6.3 months (95%CI 1.6–10.5), suggesting that a sequential rather than concurrent approach may be preferable. A currently recruiting phase III study is investigating the addition of alpelisib to maintenance trastuzumab and pertuzumab in patients with advanced *PIK3CA*-mutation-positive, HER2-positive ABC (NCT04208178). Similarly, inavolisib is undergoing phase I evaluation with maintenance trastuzumab/pertuzumab in this population (NCT03006172) and phase II evaluation in the neoadjuvant setting with trastuzumab, pertuzumab, and endocrine therapy (GeparPiPPa: NCT05306041).

#### 5.7.9. AKT1 Inhibitors

The recent success of capivasertib in ER-positive/HER2-negative advanced breast cancer has not yet led to any registered studies in HER2-positive breast cancer, but a phase I study has completed accrual to evaluate the safety of capivasertib with T-DXd in HER2-low breast cancer (NCT04556773) [113]. A second AKT1 inhibitor in development, ipatasertib, is currently undergoing phase I evaluation with maintenance pertuzumab and trastuzumab in patients with HER2-positive advanced breast cancer harbouring *PIK3CA* mutations (NCT04253561).

**Table 2 cancers-16-00023-t002:** Large randomised studies of HER2 TKIs.

Trial	Phase	Setting	Arms	Previous Treatment	*n*	ORR (%)	Median PFS (Months)	Median OS (Months)	Safety Profile
EGF100151, Geyer et al. (2006) [47]	III	≥2nd line	Lapatinib + capecitabine Capecitabine	Trastuzumab, anthracycline + taxane	324	22 14	8.4 4.1	75.0 weeks 64.7 weeks	12% vs. 11% G3 diarrhoea, 7 vs. 11% G3 PPE
Emilia, Verma et al. (2012) [38]	III	2nd line	T-DM1 Lapatinib + capecitabine (lap/cape)	100% prior trastuzumab + taxane No pertuzumab	991	44 31	9.6 6.4	29.9 25.9	41% vs. 57% all G3/4 toxicities T-DM1 higher risk of thrombocytopenia + abnormal LFTs
Nala, Saura et al. (2020) [69]	III	3rd line	Neratinib + capecitabine Lapatinib + capecitabine	75% previous T-DM1 42.5% trastuzumab + pertuzumab	621	32.8 26.7	5.6 5.5	24 22.2	24% vs. 12.5% G3 diarrhoea despite loperamide prophylaxis
HER2CLIMB, Murthy et al. (2020) [82]	II	3rd line	Trastuzumab + capecitabine + tucatinib Trastuzumab + capecitabine + placebo	100% previous T-DM1 + trastuzumab/ pertuzumab	612	40.6 22.8	7.8 5.68	24.7 19.2	12.9% vs. 8.5% G3 diarrhoea with no loperamide prophylaxis 5% vs. 0.5% G3 transaminitis
Phoebe, Xu et al. (2021 INTERIM) [104]	III	3rd line	Pyrotinib + capecitabine Lapatinib + capecitabine	100% received trastuzumab + taxane Max 2 lines of chemo	267	67.2 51.5	12.5 5.6	NR NR	30.6% vs. 8.3% G3 diarrhoea
EFG104900, Blackwell et al. (2010) [43]	III	>3rd line	Lapatinib + trastuzumab Lapatinib	100% prior trastuzumab No T-DM1 or pertuzumab	296	10.3 6.9	12.1 weeks 8.1 weeks	14 9.5	Asymptomatic cardiac events: 3.4% vs. 1.4% Symptomatic: 2% vs. 0.7%

**Table 3 cancers-16-00023-t003:** Phase II/III clinical trials in patients with HER2+ advanced breast cancer with CNS metastases.

Trial	Phase	Arms	Previous Treatment of Brain Mets	Previous Systemic Treatment	N with CNS Disease	CNS ORR (%)	CNS Median PFS (Months)	CNS Median OS (Months)
EGF105084, Lin et al. (2009) [56]	II	Lapatinib Lapatinib + capecitabine (expansion phase)	100% prior cranial radiotherapy	100% trastuzumab	240 50	6 20	2.4 3.65	6.37 NR
EGF107671, Lin et al. (2011) [57]	II	Lapatinib + capecitabine Lapatinib + topetecan	100% prior cranial radiotherapy	100% trastuzumab	22	38 0	NR NR	NR NR
Landscape, Bachelot et al. (2013) [58]	II (single arm)	Lapatinib + capecitabine	Nil	93% trastuzumab-based chemotherapy	45	65.9	5.5	17
TBCRC 022, Freedman et al. (2019) [74]	II (single arm)	Neratinib + capecitabine	92% prior neurosurgery +/− radiotherapy	Cohort A: 50% exposed to lapatinib Cohort B: 50% lapatinib naïve	49	49 33	5.5 3.1	13.3 15.1
HER2CLIMB, Murthy et al. (2020) [82]	II	Tucatinib + capecitabine + trastuzumab Placebo + capecitabine + trastuzumab	43% prior cranial radiotherapy	100% trastuzumab, pertuzumab and T-DM1	291	47.3 20	9.9 4.2	18.1 12
Kamilla, Montemuro et al., 2020, Wuerstlein et al. (2022) [38]	IIIb	T-DM1	46.8% prior cranial radiotherapy	100% trastuzumab and taxane chemotherapy	126 (AS)	21.4%	5.5	18.9
DESTINY-BREAST 01, Modi et al., 2021 [22]	II	Trastuzumab Deruxtecan	58.3% prior cranial radiotherapy 12.5% prior surgery and RT 4.2% prior surgery, RT and capecitabine	100% trastuzumab and taxane chemotherapy	24	58.3	18.1	NR
TUXEDO 1 Bartsch et al., 2021 [27])	II	Trastuzumab Deruxtecan	60% prior cranial radiotherapy	100% trastuzumab and taxane chemotherapy	15	83.3%	14	NR
DESTINY BREAST 03, Cortes et al., 2022 [24]	III	Trastuzumab Deruxtecan	N/A	100% trastuzumab and taxane chemotherapy	82	67.4%	15	NR
PATRICIA, Lin et al., 2021 [102]	II	High-dose trastuzumab/pertuzumab	100% prior cranial radiotherapy	100% trastuzumab and taxane chemotherapy	39	11%	N/A	N/A

**Table 4 cancers-16-00023-t004:** Selected novel combination trials recruiting in HER2+ ABC.

Trial Identifier	Phase	Treatment Setting	Population	Planned Sample Size	Treatment Arms	Primary Endpoint	Study Status
**Tucatinib**							
NCT03975647 (HER2CLIMB-02)	III (RCT double blind)	2nd line	HER2 + MBC	460	Tucatinib + T-DM1 vs. placebo and T-DM1	PFS	Recruited, results awaited
NCT04539938 (HER2CLIMB-04)	II (single arm)	3rd line	HER2 + MBC	70	Tucatinib + T-Dxd	ORR	Recruiting
NCT05132582 (HER2-CLIMB-05)	III	2nd line	HER2 + MBC	650	Trastuzumab and pertuzumab and Placebo Vs	PFS	Recruiting
NCT03054363	Ib/II (single arm)	≥3rd line	HR + HER2 + MBC	42	Tucatinib, palbociclib + letrozole	Phase Ib: incidence of toxicity Phase II: PFS	Active, not recruiting
NCT03501979	II (single arm)	No previous LMD specific therapy	HER2 + MBC with leptomeningeal disease	30	Tucatinib, trastuzumab + capecitabine	OS	Active, not recruiting
NCT04760431 (HER2BRAIN)	II (RCT double bind)	2nd line (progression on/after trastuzumab)	HER2 + MBC with active brain metastases	120	Trastuzumab, taxane + pertuzumab vs. trastuzumab, taxane + TKI (Tucatinib, pyrotinib or neratinib)	CNS ORR	Not yet recruiting
**Neratinib**							
NCT03377387	Ib/II	Ib: any line II: ≥2nd line	HER2 + MBC	48	Capecitabine 7/7 + neratinib	Phase Ib: MTD	Active, not recruiting
NCT01494662	II (4 cohorts, non-randomised)	Varied	HER2 + MBC with brain metastases	168	(1) Neratinib (2) Neratinib + surgical resection (3) Neratinib + capecitabine (4) Neratinib + T-DM1	CNS ORR	Active, not recruiting
**Pyrotinib**							
NCT03080805 (PHOEBE)	III (RCT)	≥2nd line (must have had prior trastuzumab + taxane)	HER2 + MBC	240	Pyrotinib + capecitabine vs. Lapatinib + capecitabine	PFS	Unknown
**Poziotinib**							
NCT02659514	II (single arm)	3rd line (must have had previous trastuzumab and T-DM1)	HER2 + MBC	67	Poziotinib monotherapy	ORR	Active, not recruiting
**Antibody Drug Conjugates**							
NCT04538742 (DESTINY-BREAST07)	Ib/II	1st line	HER2 + MBC including those with active brain metastases	350	7 cohorts combining T-DXd with: (1) Durvalumab (2) Pertuzumab (3) Paclitaxel (4) Durvalumab/ paclitaxel (5) T-DXd alone (6) Tucatinib (7) Tucatinib in active brain mets (8) T-DXd alone in active brain mets	Phase Ib/II: incidence of toxicity	Active, not recruiting
NCT04739761 (DESTINY-BREAST12)	III	≥2nd line	HER2 + MBC including those with active brain metastases		T-DXd	ORR, PFS	Active, not recruiting
NCT04492488	I/II	≥2nd line	HER2 + advanced cancers	129	MRG002	NTD, RP2D, ORR and Aes	Recruiting
NCT03944499	I	≥2nd line	HER2 + advanced cancers	297	FS-1502	NTD, RP2D, ORR and Aes	Recruiting
NCT03953833	I	≥2nd line	HER2 + advanced cancers	45	B003	NTD, RP2D, ORR and Aes	Active, not recruiting
NCT04450732	I	≥2nd line	HER2 + advanced cancers	96	GQ1001	NTD, RP2D, ORR and Aes	Recruiting

**Table 5 cancers-16-00023-t005:** Studies assessing the combination of checkpoint inhibitors with anti-HER2 agents.

Trial	Phase	Patient Number	Arms	Results/Status
PANACEA (NCT02129556)	Ib-II	*n* = 58	Trastuzumab + pembrolizumab	ORR: 15% of PD-L1 + pts No ORs among PD-L1 − pts mPFS: 2.7 mos (90% CI 2.6–4.0) in PD-L1+ mPFS: 2.5 mos (90% CI 1.4–2.7) in PD-L1-ve
KATE-2	II	*n* = 202	TDM-1 + atezolizumab TDM-1 + placebo	mPFS in PDL1 + pts T-DM1+ atezo 8.5 m T-DM1+ placebo 4.1 m HR 0.60 (CI 0.32–1.11)
KATE-3 (NCT04740918)	III	*n* = 320	TDM1+ atezolizumab/placebo	Active–not recruiting
NCT03125928	II	*n* = 50	Single arm Paclitaxel-trastuzumab-pertuzumab-atezolizumab	Recruiting
NRG-BR004 (NCT03199885)	III	*n* = 600	Paclitaxel-trastuzumab-pertuzumab-atezolizumab/placebo	Active–not recruiting

## 6. Conclusions

The recent advances and emergence of novel anti-HER2-targeted therapies has dramatically changed the outlook for patients with HER2-positive ABC, but also presented new therapeutic challenges for the practising clinician. The new HER2 ADC, T-DXd, and novel TKIs, such as tucatinib, have demonstrated their efficacy and have become standard treatments.

The optimal sequence of HER2-directed therapies will continue to change due to the dynamic nature of current efforts to expand our anti-HER2 drug armamentarium. At present, T-DXd is the standard second-line therapy, superseding T-DM1. In the third line setting (and potentially earlier in the setting of predominant CNS metastases), tucatinib, capecitabine, and trastuzumab is the new standard option, where available, based on the results of the HER2CLIMB trial, although the use of capecitabine/neratinib (or if necessary capecitabine/lapatinib) should be considered if tucatinib is unavailable. For patients relapsing soon after adjuvant T-DM1, T-DXd should be considered as a first-line therapy for metastatic disease, although tucatinib, capecitabine, and trastuzumab would currently be preferred for brain-only relapses in this setting.

The management of CNS disease has progressed significantly with the next generation of HER2 TKIs, with the improved efficacy seen with both neratinib and tucatinib and the favourable toxicity profile of the latter, and, more recently, accumulating evidence that the unprecedented efficacy of T-DXd also extends to the CNS. Current studies evaluating these agents in earlier lines of therapy for ABC and in high-risk early breast cancer will determine whether they can prevent CNS disease and become the ultimate panacea for HER2-positive breast cancer.

## Figures and Tables

**Figure 1 cancers-16-00023-f001:**
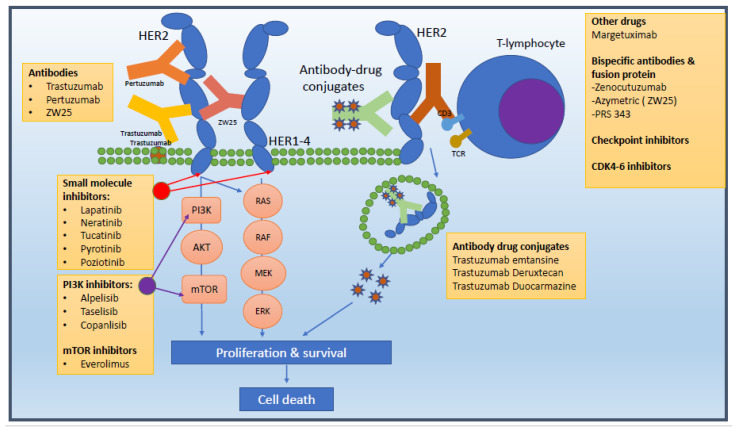
HER2 signalling cascade and drug targets.

**Table 1 cancers-16-00023-t001:** Pivotal first- and second-line clinical trials in HER2+ advanced breast cancer.

Trial	Phase	Patient Number /Line	Arms	Median PFS (Months)	Median OS (Months)
Cleopatra	III	*n* = 808 1st line	Docetaxel + trastuzumab + pertuzumab Docetaxel + trastuzumab + placebo	18.5 12.4	57.1 40.8
Peruse	IIIb	*n* = 1436 1st line	Docetaxel + pertuzumab + trastuzumab Paclitaxel + pertuzumab + trastuzumab Nab-paclitaxel + pertuzumab + trastuzumab	19.4 23.2 19.2	66.5 64 70.9
Pertain	II	*n* = 129 1st line	Aromatase inhibitor or taxane + Trastuzumab + pertuzumab Aromatase inhibitor or taxane + Trastuzumab	21 16	60 57
Marianne	III	*n* = 1095 First line	Taxane + trastuzumab TDM-1 TDM-1 + pertuzumab	13.7 14.1 15.2	50.9 53.7 51.8
Emilia	III	*n* = 991 Second line	TDM-1 Capecitabine + lapatinib	9.6 6.4	29.9 25.9
Destiny Breast 03	III	*n* = 699 Second line	Trastuzumab deruxtecan T-DM1	28.8 6.8	NR NR

PFS = progression-free survival; OS = overall survival.

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
