# Peer review of "Systemic Therapies for HER2-Positive Advanced Breast Cancer"

_cancers, 2023, doi:10.3390/cancers16010023_

Round 1
Reviewer 1 Report
Comments and Suggestions for Authors
In this manuscript, the authors have described the current anti-HER2 therapies and discussed treatment strategies for patients with HER2-positive advanced breast cancer. The review article summarized many studies. However, there are some major concerns, which should be addressed or clarified prior to publication. I am sorry, but the manuscript should be revised before being accepted for publication.
The authors have reported a lot of information that should be better schematized in tables to help the readers to understand the research data and to follow the flow of information.
For instance, a specific table for each line of therapy should be provided by the authors.
I suggest the authors update the data on studies related to T-DM1(von Minckwitz, et al, 2019; Denkert C et al, 2023).
Please carefully revise the table 1, 2 and 3, and check if they are correctly mentioned in the main manuscript. There’s some confusion. Table 3 is not mentioned in the main text.
Line 82: Could the authors explain how HER2/HER3 are the most powerful heterodimers.
Figure 1 is missing.
Line 91. The CLEOPATRA study is not reported in table 1. The table 1contains the “Large randomised studies of HER2 TKIs”.
I suggest the authors include a scheme that graphically illustrates the mechanism of action of anti-HER2 ADCs.
It would be helpful to include another scheme relative to the TKls mechanism of action.
Given the contribution of the immune system to anticancer therapy, the authors should provide a table, which clearly and concisely displays information on the combination of immune checkpoint with anti-HER2 antibodies.
Line 460. It may be useful in this context to mention the study of Hurvitz S.A et al, 2022
The authors should deeply comment on the mechanisms of antibody-dependent cell-mediated cytotoxicity.
The authors should provide a table displaying information on phase II/III trials of PI3K/AKT/mTOR inhibitors.
Comments on the Quality of English Language
This manuscript requires a few typos or grammatical corrections. There are many frequent punctuation errors.
Author Response
In this manuscript, the authors have described the current anti-HER2 therapies and discussed treatment strategies for patients with HER2-positive advanced breast cancer. The review article summarized many studies. However, there are some major concerns, which should be addressed or clarified prior to publication. I am sorry, but the manuscript should be revised before being accepted for publication.
The authors have reported a lot of information that should be better schematized in tables to help the readers to understand the research data and to follow the flow of information.
For instance, a specific table for each line of therapy should be provided by the authors.
Thank you for your comments. A new table (Table 1) has been included to address your comments. As it is not possible to include all available studies for each line of treatment, we have included the key studies for 1st and 2nd line treatment to facilitate the reader navigate through the available data.
I suggest the authors update the data on studies related to T-DM1(von Minckwitz, et al, 2019; Denkert C et al, 2023).
Thank you for your comments. The studies by von Minckwitz et al 2019 ( Trastuzumab Emtansive for Residual Invasive HER2-positive breast cancer NEJM 2019 380(7):617-628) and Denkert et al 2023 ( Biomarker data from the phase II study KATHERINE) were not included, as these both focused on early HER2 positive breast cancer.
Please carefully revise the table 1, 2 and 3, and check if they are correctly mentioned in the main manuscript. There’s some confusion. Table 3 is not mentioned in the main text.
Thank you for pointing this out, we have now included reference in the main text for each of the tables.
Line 82: Could the authors explain how HER2/HER3 are the most powerful heterodimers.
Thank you for your comment. This sentence has now been adjusted to explain this further, whilst a suitable reference to offer further insight to this statement has been added.
Figure 1 is missing.
Thank you and apologies for this omission. Figure 1 has now been added.
Line 91. The CLEOPATRA study is not reported in table 1. The table 1contains the “Large randomised studies of HER2 TKIs”.
Thank you for this comment. This has now been addressed by the introduction of the new table 1 that summaries pivotal studies in management of HER2+ breast cancer.
I suggest the authors include a scheme that graphically illustrates the mechanism of action of anti-HER2 ADCs. It would be helpful to include another scheme relative to the TKls mechanism of action.
Thank you for your comment. This is now hopefully addressed by Figure 1.
Given the contribution of the immune system to anticancer therapy, the authors should provide a table, which clearly and concisely displays information on the combination of immune checkpoint with anti-HER2 antibodies.
Thank you for your comment. In view of the large number of tables already incorporated in this review, we instead have included a reference to the very comprehensive review by Agostinetto et al Cancers 2022 , which offers a much more details view on this topic. We hope you agree with this approach, but if you still feel strongly about this, we can add another table.
Line 460. It may be useful in this context to mention the study of Hurvitz S.A et al, 2022
Thank you for comment. We would be grateful if you could clarify which study by Dr Hurvitz you are referring to. The ASTEFANIA study by Dr Hurvitz has already been mentioned in the relevant section in the main text. Thank you and apologies if we have missed your point.
The authors should deeply comment on the mechanisms of antibody-dependent cell-mediated cytotoxicity.
Thank you for your comment. The role of ADCC as a therapeutic effect of various mAbs has been tested in preclinical models and validated in patients and is currently known to be one of the main mechanisms that contribute to the effect of trastuzumab. Although it is essential to mention this, a deep analysis of this mechanism is beyond the scope of this review.
The authors should provide a table displaying information on phase II/III trials of PI3K/AKT/mTOR inhibitors.
Thank you for your comments. These have now been incorporated in the table summarising current novel approaches in advanced HER2 positive breast cancer.
Reviewer 2 Report
Comments and Suggestions for Authors
This is a narrative review article about current systemic therapies for patients with HER2-positive ABC. They did provide lots of updated information and insights into this important field. However, the content and structure of this review could be improved before acceptance.
1. I have not found the Figure 1 file in this manuscript. Since the optimal sequence of treatment strategy has been addressed, I may suggest the authors consider adding a flowchart of Systemic Therapy Strategies in advanced HER2-positive breast cancer for reference in clinical practice.
2. Since only systemic treatment strategies have been focused on this manuscript, I may suggest revising the title accordingly.
3. All the tables should provide the last updated date as a footnote.
4. The references in this article could be more recent. For instance, the ESMO Congress 2023 is taking place these days and new information on advanced breast cancer has been released. I would recommend the authors consider update the content in your revisions.
Author Response
This is a narrative review article about current systemic therapies for patients with HER2-positive ABC. They did provide lots of updated information and insights into this important field. However, the content and structure of this review could be improved before acceptance.
- I have not found the Figure 1 file in this manuscript. Since the optimal sequence of treatment strategy has been addressed, I may suggest the authors consider adding a flowchart of Systemic Therapy Strategies in advanced HER2-positive breast cancer for reference in clinical practice.
Thank you for comments. We have now included Figure 1, which unfortunately was omitted in error and which indeed is acting as a flowchart as per your secondary comment. We hope you find this relevant and useful.
2. Since only systemic treatment strategies have been focused on this manuscript, I may suggest revising the title accordingly.
Thank you for your suggestion, the title has indeed now been amended to reflect the content appropriately.
3. All the tables should provide the last updated date as a footnote.
Thank you for your comment, tables have now been updated as per your comments.
4. The references in this article could be more recent. For instance, the ESMO Congress 2023 is taking place these days and new information on advanced breast cancer has been released. I would recommend the authors consider update the content in your revisions.
Thank you for your suggestion. Please note, this manuscript was submitted prior to ESMO 2023 and that is possibly why there have been no mention of any relevant studies presented during this conference. Although there were no ground breaking studies presented in ESMO 2023 in relation to the subject of this review, there were some interesting findings on the pooled analysis for metastatic brain disease in the DESTINY BREAST 01, 02 and 03, which were presented. These were included in the text to address your comments, thank you again.
Reviewer 3 Report
Comments and Suggestions for Authors
It is a comprehensive review of the management of ABC. but I am not an oncologist to provide any major comment
Author Response
Thank you very much for taking the time to review our manuscript. We hope that you enjoyed reading this and despite the fact you may not specialise in the management of breast cancer patients, we hope that our review gave you a taste of the ongoing clinical research in HER2 positive breast cancer.
Reviewer 4 Report
Comments and Suggestions for Authors
The reviewed manuscript is undoubtedly interesting and relevant as it highlights the current treatment of patients with HER2-positive advanced breast cancer. This remains a serious problem of oncology despite the progress in the conservative treatment methods.
The fact that this problem is a burning subject of search and study both for medical researchers and medical practitioners is quite obvious due to the numerous scientific publications. According to the PubMed data, the number of these works has exceeded 400 in the current year. Many of the articles are reviews which thoroughly and professionally depict the advances in this area of oncology as well as predict further researches. The review by Sandra M. Swain et al, presented online in 2022 and further published at the beginning of 2023 in Nature Reviews Drug Discovery, is an example of such works.
The reviewed manuscript in many aspects echoes with this article by Swain, et al, who have treated breast cancer patients for more than twenty years. This similarity decreases the novelty of the material but not the quality of its presentation. The authors scientifically discuss the current state-of-art and further development of anti-tumor drugs based on last achievements in this sphere.
In our opinion, the manuscript is of considerable interest for oncologists and mammalogists and could be accepted for publication in "Cancers" in the present form.
Author Response
Thank you very much for taking the time to review our manuscript and thank you for your positive comments.
Round 2
Reviewer 1 Report
Comments and Suggestions for Authors
The manuscript has improved after the revision. But I still suggest some comments for minor revisions.
Figure 1 is still missing. Please revise carefully.
Table numbers should be in sequential order. Please revise Table 4.
I suggested including a comprehensive review of the ASTEFANIA trial (NCT04873362). It may be helpful for the reader.
Review Future Oncol. 2022 Oct;18(32):3563-3572. doi: 10.2217/fon-2022-0485. Epub 2022 Nov 16.
There are only four tables in the main text. Could the authors provide another table with information on the combination of immune checkpoint with anti-HER2 antibodies? This table might help readers follow the flow of information.
Comments on the Quality of English LanguageMinor editing of English language required
Author Response
The manuscript has improved after the revision. But I still suggest some comments for minor revisions.
Figure 1 is still missing. Please revise carefully.
Thank you. We have now ensured Figure 1 is uploaded correctly.
Table numbers should be in sequential order. Please revise Table 4.
Thank you for your comments. The tables have been corrected and Table 4 has been revised accordingly.
I suggested including a comprehensive review of the ASTEFANIA trial (NCT04873362). It may be helpful for the reader.
Review Future Oncol. 2022 Oct;18(32):3563-3572. doi: 10.2217/fon-2022-0485. Epub 2022 Nov 16.
Thank you. The section on ASTEFANIA has been revised to include more detail.
There are only four tables in the main text. Could the authors provide another table with information on the combination of immune checkpoint with anti-HER2 antibodies? This table might help readers follow the flow of information.
Thank you. We have now included a separate table on combination of checkpoint inhibitors and anti-HER2 therapy studies.
Reviewer 2 Report
Comments and Suggestions for Authors
1. I could not find table 4 in the revised version.
2. Please make sure all clinical data are updated and corrected. After that, I would like to endorse the acceptance of your article.
Author Response
- I could not find table 4 in the revised version.
Thank you for your comment. We have now ensure Table 4 is included and clearly labelled.
2. Please make sure all clinical data are updated and corrected. After that, I would like to endorse the acceptance of your article.
Thank you for comment. We have now ensured that all relevant clinical data has been updated and included most recent updates from ESMO 2023.